# Evaluation of a New Simplified Inertial Sensor Method against Electrogoniometer for Measuring Wrist Motion in Occupational Studies

**DOI:** 10.3390/s22041690

**Published:** 2022-02-21

**Authors:** Karnica Manivasagam, Liyun Yang

**Affiliations:** 1Division of Ergonomics, School of Engineering Sciences in Chemistry, Biotechnology and Health, KTH Royal Institute of Technology, Hälsovägen 11C, SE-141 57 Huddinge, Sweden; karnica@kth.se; 2Unit of Occupational Medicine, Institute of Environmental Medicine, Karolinska Institutet, SE-171 77 Stockholm, Sweden

**Keywords:** inertial measurement units, gyroscope, goniometer, wrist flexion velocity, work-related musculoskeletal disorders, ergonomics, hand-intensive work

## Abstract

Wrist velocity is an important risk factor for work-related musculoskeletal disorders in the elbow/hand, which is also difficult to assess by observation or self-reports. This study aimed to evaluate a new convenient and low-cost inertial measurement unit (IMU)-based method using gyroscope signals against an electrogoniometer for measuring wrist flexion velocity. Twelve participants performed standard wrist movements and simulated work tasks while equipped with both systems. Two computational algorithms for the IMU-based system, i.e., IMU_norm_ and IMU_flex_, were used. For wrist flexion/extension, the mean absolute errors (MAEs) of median wrist flexion velocity compared to the goniometer were <10.1°/s for IMU_norm_ and <4.1°/s for IMU_flex_. During wrist deviation and pronation/supination, all methods showed errors, where the IMU_norm_ method had the largest overestimations. For simulated work tasks, the IMU_flex_ method had small bias and better accuracy than the IMU_norm_ method compared to the goniometer, with the MAEs of median wrist flexion velocity <5.8°/s. The results suggest that the IMU-based method can be considered as a convenient method to assess wrist motion for occupational studies or ergonomic evaluations for the design of workstations and tools by both researchers and practitioners, and the IMU_flex_ method is preferred. Future studies need to examine algorithms to further improve the accuracy of the IMU-based method in tasks of larger variations, as well as easy calibration procedures.

## 1. Introduction

Work-related musculoskeletal disorders (WMSDs) of the hand/wrist are associated with relatively high medical costs and loss of work days [1]. Repetitive manual work, forceful exertions, prolonged nonneutral postures of the wrist, intensive wrist movements, and hand-arm vibration are identified as critical risk factors for hand/wrist WMSDs, which are common in service industries, manufacturing industries, office work, as well as the healthcare sector [2,3,4,5,6,7,8]. From a meta-analysis on the prevalence of WMSDs in Europe’s secondary industries during the 21st century, wrist WMSD is among the most common WMSDs with a twelve-month prevalence of 42% [9].

There are a few observational methods that consider hand/wrist workload and its associated risk [10]. However, the micro-postures and small body parts, including hand/wrist, are shown to be difficult to assess by observations with satisfactory accuracy [10,11]. Poor correlations were reported between the raters using the same observational method on the upper limb, which included the risk factors of frequency and wrist posture [12]. Poor correlations were also found between different observational methods in estimating the hand repetitiveness [13]. Technical measurement methods can provide a quantitative risk assessment with good accuracy and reliability, which may contribute to the establishment of exposure–response relationships [14,15,16]. There are many IMU systems developed for occupational exposure assessment [17]. However, most multi-sensor systems are complex to use in the field by practitioners [18,19]. In addition, many systems can be used for assessments of the upper arm and trunk, but few can be used for assessment of wrist motion [20,21,22]. Recent research has explored a multi-sensor system focusing on wrist risk assessment. For example, Seidel and colleagues used a multi-sensor system including inertial sensors, a potentiometer, an electromyograph, and a data logger to quantify the hand activity level, showing good accuracy [23]. Still, the wear burden, the required expertise, and the demand of monetary and time resources for such multi-sensor systems can be relatively high.

The importance of assessing wrist angular velocity has been identified and raised by a few authors [24,25,26,27]. Researchers could establish a quantitative exposure–response relationship between the wrist flexion velocity and the disorders in the hand/wrist, and an action limit of 20°/s for median wrist velocity during an 8 h working day was proposed [25,28]. The electrogoniometer is one commonly used and validated technical measurement method for assessing wrist motion in occupational studies [29]. The proposed action limit on median wrist velocity was also based on measurement data of about 50 occupational groups using electrogoniometers [25]. Although being widely used in research, the electrogoniometer has a few disadvantages of being fragile, expensive, bringing certain burdens to the wearers as it requires cables and a logger to be carried, as well as requiring resources for data analysis.

New generations of Inertial Measurement Units (IMUs) with built-in gyroscopes and Bluetooth technology provide potentials of new methods for assessing wrist velocity [30]. The IMUs can be connected wirelessly to a smartphone application to process the data and generate assessment results automatically, which are convenient for both the wearer, practitioners, and researchers, and at the same time being cheaper and less resource-demanding for data analysis [31,32]. Common IMU systems usually integrate the gyroscopes’ and accelerometers’ signals to improve the performance of posture and motion assessment under rapid movements [33] where the drift in the gyroscope would be a problem over time and the integration of a magnetometer to correct the drift would be affected by the local magnetic field [34]. It is possible to use the gyroscope directly for velocity assessment, where the gyroscope drift that originates from integration and accumulation of noise will not be a concern. Before such a gyroscope and smartphone application method to be used in the field, the performance of the IMU-based method with various algorithms needs to be evaluated. As the goniometer has been widely used in the field and has laid the foundation for the proposed action limit on wrist velocity at work, it is considered as a standard measure for evaluating the IMU-based method.

Therefore, the aim of this study was to evaluate a new simplified IMU-based method, with two computational algorithms, i.e., the IMU_norm_ and the IMU_flex_, against an electrogoniometer for measuring wrist flexion velocity during standard wrist movements and simulated work tasks.

## 2. Methods

### 2.1. Participants

Twelve participants (six males and six females, all right-handed) volunteered in the study. The inclusion criteria were healthy adults and pain-free in the dominant wrist/hand. Participants were provided with information about the purpose and design of the study, and signed an informed consent ahead of the experiment. The mean (SD) age of the participants was 26 years (4.1), with a mean height of 169 cm (7.3), mean weight of 67 kg (14.9), and mean body mass index of 23.2 (4.3). The study was approved by the Regional Ethics Committee in Stockholm (Dnr, 2019-01206).

### 2.2. Measurement Systems

A twin axis electrical goniometer (Biometrics Ltd., Newport, UK) with a sampling frequency of 20 Hz was used as the standard for measuring wrist velocity. The goniometer, with a sampling frequency 52 Hz, was placed on the dorsal surface of the dominant hand with one end-block on the third metacarpal of the hand and the other on the forearm midline, at a distance from the wrist joint where the cables between the two end-blocks were kept straight and not squeezed when the wrist was fully extended. The participants were instructed to hold their arms flat on a table for calibrating the neutral wrist position at the beginning of the experiment (Figure 1). 

Two IMU sensors (Movesense, Suunto, Helsinki) were placed at the top of goniometer end-blocks on both the forearm and the hand (Figure 1a,b). The IMUs were attached using double-sided tape beneath and a medical tape on top to ensure fixation and avoid relative movements. Additionally, a customized wooden block was mounted between the hand IMU and the goniometer end-block for better stability due to the triangle shape of the end-block on the hand. The IMUs were connected and recorded using an open-source iPhone application Movesense showcase v.1.0.5 (Amer Sports Digital Services Oy, Helsinki). 

### 2.3. Experimental Design

Before the experiment started, the participants were informed about the tasks that they would perform and were given the opportunity to test and familiarize themselves with the wrist movements. Two types of tests were included in the study to evaluate the IMU-based method against the goniometer: (1) standard wrist movements in three planes, i.e., flexion/extension, radial/ulnar deviation, and pronation/supination, at paces of 30, 60, and 90 beats per minute (bpm), following a metronome (Figure 2a–c); and (2) simulated work tasks, which involved different types of hand/wrist movements: blow-drying hair, folding paper planes, and sorting mail (Figure 2d–f). Each standard wrist movement was performed for at least 10 cycles and all in a sitting position by a desk. Each work task was performed for one minute at the participants’ own chosen pace. The blow-drying hair and sorting mail were performed in a standing position, whilst folding paper planes was performed in a sitting position. 

### 2.4. Signal Processing and Statistical Analysis

The measurement data from both the systems were processed in MATLAB (2021, MathWorks, Inc., Natick, MA, USA). For the goniometer, the flexion/extension angle output was low-pass-filtered at 5 Hz [35] and the derivatives of the flexion/extension angle were calculated as wrist flexion velocity. For the IMUs, the output from the gyroscopes in three axes of both IMUs on the hand and forearm were synchronized, low-pass-filtered (4th-order Butterworth filter with a cut-off frequency at 5 Hz), and down-sampled to 20 Hz. Two computational algorithms were used for calculating the wrist flexion velocity: IMU_norm_ and IMU_flex_. For IMU_norm_, the vector norm of gyroscopes of both the hand and forearm IMUs were calculated and then subtracted, and the absolute value of the difference was taken as the wrist velocity as follows: (1)$$v_{wrist}=\left|\sqrt{\;{\left(g_{xhand}\right)}^{2}+{\left(g_{yhand}\right)}^{2}+{\left(g_{zhand}\right)}^{2}}-\sqrt{{\left(g_{xforearm}\right)}^{2}+{\left(g_{yforearm}\right)}^{2}+{\left(g_{zforearm}\right)}^{2}}\right|$$

For IMU_flex_, the gyroscope output at the x-axis (i.e., the flexion/extension axis) of the hand and forearm IMUs were subtracted as the wrist velocity as follows:(2)$$v_{wrist}=\sqrt{\;{\left(g_{xhand}-g_{xforearm}\right)}^{2}\;}$$

The calculated IMU-based wrist velocities were synchronized with the goniometer output, and the 10th, 50th, and 90th percentiles of the wrist flexion velocity were calculated for each of the standard movements and simulated work tasks. The two computational algorithms for IMUs were compared against the goniometer separately. The mean absolute errors (MAEs) and its standard deviation (SD) were computed between each pair of comparison for all tasks at the 10th, 50th, and 90th percentile. In addition, correlation plots and Bland–Altman plots of the median wrist flexion velocity for all tasks were used to compare the IMU_norm_ and IMU_flex_ against the goniometer.

## 3. Results

The measured wrist flexion velocity by the goniometer and two IMU-based methods and the mean absolute errors (MAEs) from the comparison during standard wrist movements are shown in Table 1. For the wrist flexion/extension, the IMU_flex_ had good accuracy with the MAEs ranging from 1.4°/s to 4.1°/s for median wrist velocity as the pace increased. The IMU_norm_ had slightly larger errors with the MAEs of median wrist velocity ranging from 4.1°/s up to 10.1°/s. Larger errors were observed for the 90th percentile of wrist velocity between the IMU-based methods and goniometer as the speed increased. The correlation and limits of agreement between the IMU_norm_ and goniometer during wrist flexion/extension are shown in Figure 3a,b, and those between the IMU_flex_ and goniometer are shown in Figure 3c,d. The squared correlation coefficient (r^2^) was high between both IMU-based methods and the goniometer (r^2^ = 0.98 and 1.00, respectively). The IMU_norm_ had a tendency of underestimation when the flexion speed increased from 30 to 90 BPM, with limits of agreement of −20°/s and 6.4°/s.

During wrist deviation and pronation/supination, the flexion velocity should theoretically be close to zero. However, errors were observed in the goniometer and two IMU-based methods, where the IMU_norm_ had the largest overestimations with the mean values of the 50th percentile of flexion velocity up to 23.4°/s for deviation at 90 BPM and 51.4°/s for pronation/supination at 90 BPM (see Individual Method, Table 1). The correlation and limits of agreement between the IMU_norm_ and goniometer during wrist deviation are shown in Figure 4a,b, and those during pronation/supination are shown in Figure 5a,b. Distinctive differences were observed between the IMU_norm_ and goniometer, with larger differences as the speed of motion increased from 30 to 90 BPM, and larger differences during the pronation/supination than deviation. The IMU_flex_ and goniometer had errors calculating the wrist flexion velocity at similar levels during deviation, with mean values of the 50th percentile wrist velocity up to 8.1°/s and 7.9°/s, respectively (see Individual Method, Table 1). For wrist pronation/supination, the IMU_flex_ had smaller errors compared to the goniometer (mean values of 50th percentile wrist velocity up to 9.6°/s vs. 19.1°/s, respectively). The correlation and limits of agreement between the IMU_flex_ and goniometer during wrist deviation are shown in Figure 4c,d, and those during pronation/supination are shown in Figure 5c,d. As the values of the goniometer during deviation and pronation/supination were shown to be flawed, the difference between the IMU-based methods and the goniometer in these two wrist movements should not be considered as errors compared to the ground truth.

The wrist flexion velocity measured by the goniometer and two IMU-based methods and the mean absolute errors (MAEs) during simulated work tasks are shown in Table 2. The IMU_flex_ had an overall better accuracy than the IMU_norm_ compared to the goniometer, with the MAEs of median wrist flexion velocity <5.8°/s for all work tasks, vs. <10.6°/s for IMU_norm_ (see Comparison of Methods, Table 2). The largest MAEs were observed during blow-drying hair for both IMU-based methods, whereas the smallest differences were observed during folding paper planes. The correlation and limits of agreement between the IMU_norm_ and goniometer during three work tasks are shown in Figure 6a,b, and those between the IMU_flex_ and goniometer are shown in Figure 6c,d. The IMU_flex_ had a high squared correlation coefficient compared to the goniometer (r^2^ = 0.93), a small bias of 0.78°/s, and limits of agreement of −10°/s and 12°/s. The IMU_norm_ had a lower squared correlation coefficient compared to the goniometer (r^2^ = 0.71), a bias of −3°/s, and larger limits of agreement of −26°/s and 20°/s.

## 4. Discussion

This study evaluated a new simplified IMU-based method using two computational algorithms, i.e., the IMU_norm_ and the IMU_flex_, against the electrogoniometer for measuring wrist flexion velocity during standard wrist movements and simulated work tasks. The results showed that the IMU_flex_ method had an overall better performance than the IMU_norm_ compared to the goniometer in all standard wrist movements and simulated work tasks, with relatively small bias. The new IMU-based method has a great potential to be applied as a convenient method for risk assessment of wrist motion at work by both researchers and practitioners.

The standard wrist movements including motion in three planes were chosen to test the performance of the IMU-based method compared to the goniometer under constrained conditions. During standard wrist flexion/extension movements, it was observed that the IMU_flex_ had high agreement and small bias compared with the goniometer method (Table 1 and Figure 3). The IMU_norm_ method had slightly larger differences compared to the goniometer, especially when the speed increased from 30 to 90 BPM.

For the radial-ulnar deviation movements, the wrist flexion velocity should theoretically be close to zero. However, errors were observed both by the goniometer and the IMU-based methods. The measured flexion velocity, which deviated from zero by all methods, could partly be explained by the coupling of wrist flexion-extension and deviation [36]. Therefore, there were wrist movements in the flexion-extension plane even though the participants were instructed to only move in the radial-ulnar deviation plane, which are mainly due to the constraints of carpal ligaments and muscle contraction patterns [36]. In addition, the errors of the IMU-based methods could be caused by the unaligned deviation and rotation axes of the built-in gyroscopes of the hand and forearm IMUs, and the soft tissue artifacts, i.e., the relative movement between the skin and underlying bone. For the IMU_norm_ method, the extra high errors were partly expected (Table 1 and Figure 4 and Figure 5), as the inclusion of all three axes of the gyroscopes would also record the deviation and rotation movements of the wrist. For the IMU_flex_ method, as the flexion axes of the gyroscopes of the hand and forearm IMUs were assumed to be aligned, estimation errors would occur when the axes were not perfectly aligned. The misalignment could happen due to the irregular articular surface of the carpal bones, the placement of the two IMUs on goniometer end-blocks, and combined wrist movements on more than two planes. For the goniometer, the errors can be explained by its inherent crosstalk between the flexion and deviation recordings due to twisting of the goniometer transducer, the impact of forearm rotation, and the movement of skin [35,37,38], which could be observed during the deviation and supination/pronation movements (see Individual method, Table 1). Nevertheless, the IMU_flex_ method showed small bias in the wrist flexion velocity results compared with the goniometer during deviation (Table 1 and Figure 4c,d). For wrist pronation/supination, the IMU_flex_ had smaller errors compared to the goniometer, i.e., smaller values above zero in this case (Table 1 and Figure 5c,d).

Three tasks using the hand/wrist in different intensities were included in the experiment. The task of folding paper planes was the least intensive and showed the smallest wrist velocities measured by all three methods. It also had the smallest differences between the two IMU-based methods and goniometer (see Comparison of methods, Table 2). Blow-drying hair and sorting mail involved more intensive wrist motions and combined motions on different planes of the wrist joint. The task of sorting mail showed the highest wrist velocities measured by all three methods, with the mean 50th percentile of wrist velocity between 45.6 and 51.8°/s (see Individual method, Table 2). The task of blow-drying hair had a larger variation between the participants, as they were instructed to perform the task at their own chosen pace. Overall, the IMU_flex_ method had smaller MAEs than the IMU_norm_ in comparison with the goniometer during all simulated work tasks.

### Limitations and Future Studies

As discussed above, during the radial/ulnar deviation and pronation/supination movements, the recorded wrist flexion velocity should theoretically be close to zero. Nonetheless, the goniometer, which is considered as a standard measure in this study, showed errors. Therefore, the difference between the goniometer and two IMU-based methods cannot be considered as errors compared to the ground truth. Still, the crosstalk was shown to have a marginal effect on the assessment when the wrist flexion summary measures were used in the occupational studies [37,39]. In addition, to be used for risk assessment in occupational studies with the proposed action level of wrist velocity [28], the comparison of the IMU-based methods against the goniometer is still valuable. Future work will evaluate both the IMU-based methods and the goniometer against an optical motion tracking system, which can be considered as a standard measurement in a laboratory environment.

Another limitation was that in order to equip both systems on participants, due to lack of space on the hand and forearm, the IMUs were mounted on the goniometer end-block, which was not ideal. When used by itself for measuring wrist motion, the IMUs would be placed on the skin, and the forearm IMU would be closer to the wrist joint. In this way, the two IMUs can be better aligned, and the impact of soft tissue artifacts can be reduced, especially during pronation/supination.

In the current study, no extra calibration procedure was performed to align the gyroscope axes, as the IMUs were mounted on the hand and forearm so the text on both IMU cases were aligned. The two IMU-based algorithms could be compared to a goniometer and showed sufficient accuracy in simulated work tasks. One could expect that when using in the field studies, the two IMUs may be misaligned due to errors or the relative movement of sensors on the skin, and calibration would be crucial to reduce the errors. Therefore, simple calibration procedures need to be explored and evaluated for future field studies in the next step.

In addition, in the field where exposures of a whole working day are measured, the variation in the hand/wrist movements will be higher compared to the laboratory setting. The sample size of this study was small, but for similar studies on wrist motion with human participants, a sample size of 10 to 12 was common [36,40].

Future studies need to examine algorithms to improve the accuracy of the IMU-based method, especially in tasks involving combined movements of wrist deviation and pronation/supination and movements of larger variation. Potential calibration procedures can be explored to improve the alignment of gyroscope axes of the two IMUs, without adding too much complexity to the method for being used by practitioners. It is also of great interest to investigate the performance of the IMU-based method in field studies. When the accuracy is of highest need, sophisticated measurement methods such as an optical motion tracking system can be used in the laboratory, while for occupational studies in the field, or for studies evaluating new work designs and tools involving hand-intensive work, the IMU-based method has the great advantage of being easy to use, having a low wearer burden, being cost-efficient, and providing direct risk assessment results.

## 5. Conclusions

A new simplified IMU-based method using two computational algorithms, i.e., the IMU_norm_ and the IMU_flex_, was evaluated against the goniometer for measuring wrist flexion velocity in this study. The IMU_flex_ method showed small bias in small- to medium-paced standard hand/wrist movements and all simulated work tasks compared to the goniometer. The results suggest that the IMU-based method has great potential to be used for risk assessments of wrist motion by both researchers and practitioners, in occupational studies, and in design and evaluating new workstations or tools. Future work should look into algorithms to improve the accuracy of the IMU-based method in tasks involving combined movements of wrist deviation and pronation/supination, as well as easy calibration procedures for field studies.

## Figures and Tables

**Figure 1 sensors-22-01690-f001:**
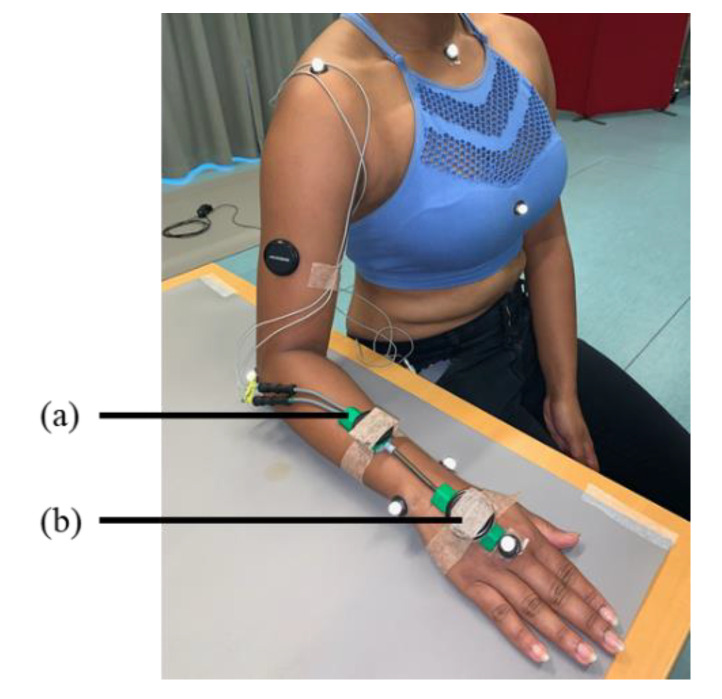
Placement of the goniometer and the two inertial measurement units (IMUs), mounted on the goniometer end-blocks, on (**a**) the forearm close to the wrist, and (**b**) the middle of the hand.

**Figure 2 sensors-22-01690-f002:**
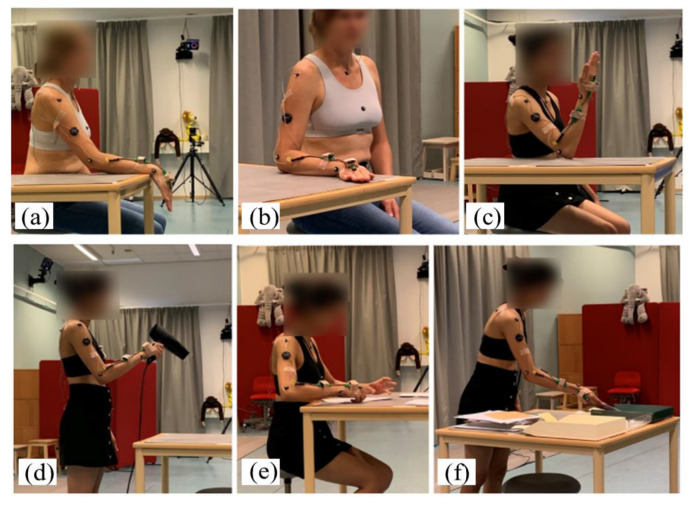
Three standard wrist movements including: (**a**) flexion/extension, (**b**) radial/ulnar deviation, and (**c**) pronation/supination; and three simulated work tasks including: (**d**) blow-drying hair, (**e**) folding paper planes, and (**f**) sorting mail.

**Figure 3 sensors-22-01690-f003:**
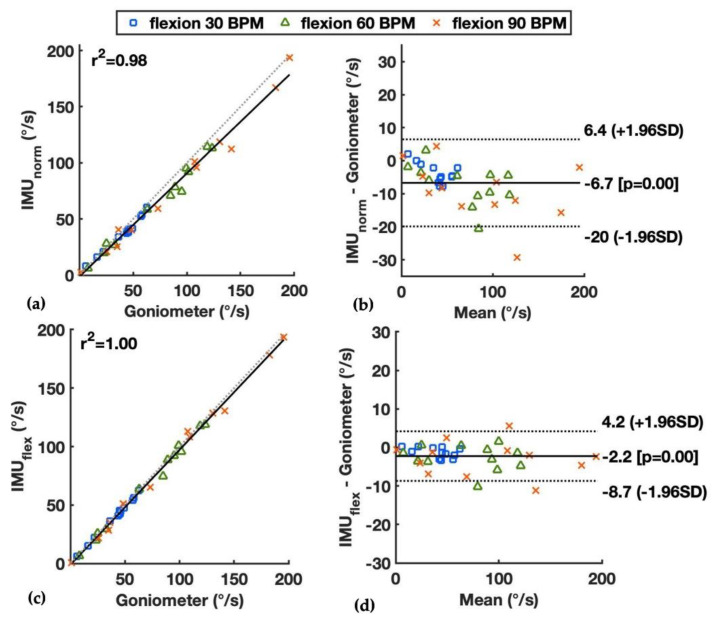
Median wrist flexion velocity during wrist flexion/extension at paces of 30, 60, and 90 BPM measured by the goniometer and the IMU-based methods, showing by linear correlation and Bland–Altman plots: (**a**,**b**) the IMU_norm_ and goniometer; and (**c**,**d**) the IMU_flex_ and goniometer. Squared correlation coefficients (r^2^) and limits of agreements are presented.

**Figure 4 sensors-22-01690-f004:**
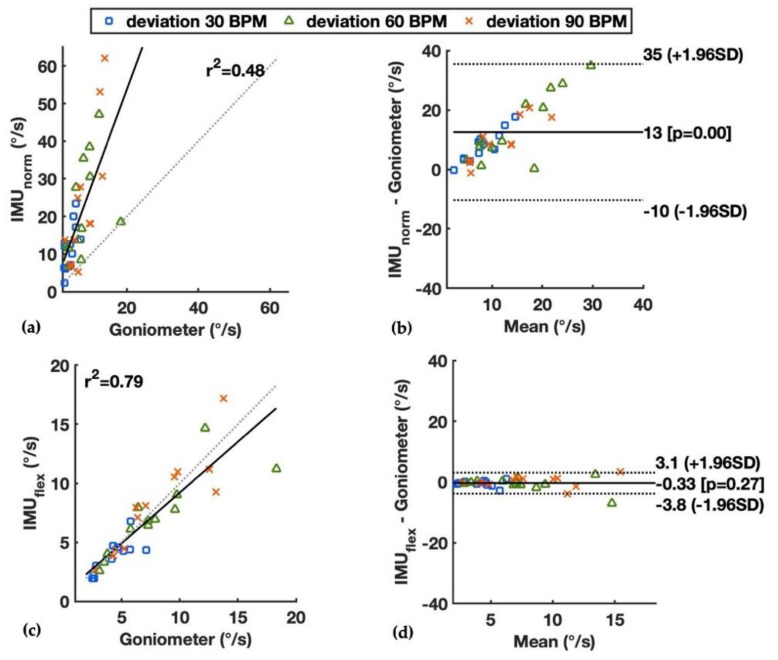
Median wrist flexion velocity during wrist deviation at paces of 30, 60, and 90 BPM measured by the goniometer and the IMU-based methods, showing by linear correlation and Bland–Altman plots: (**a**,**b**) the IMU_norm_ and goniometer; and (**c**,**d**) the IMU_flex_ and goniometer. Squared correlation coefficients (r^2^) and limits of agreements are presented.

**Figure 5 sensors-22-01690-f005:**
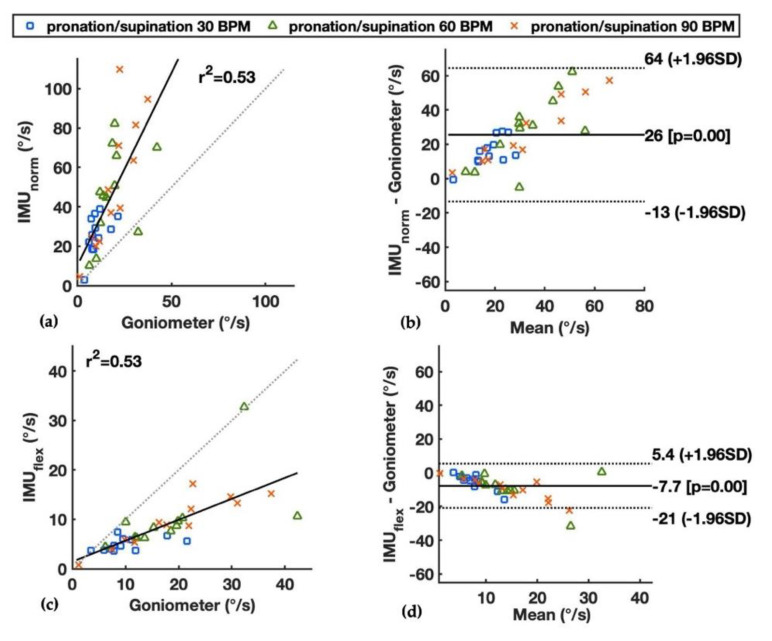
Median wrist flexion velocity during wrist pronation/supination at paces of 30, 60, and 90 BPM measured by the goniometer and the IMU-based methods, showing by linear correlation and Bland–Altman plots: (**a**,**b**) the IMU_norm_ and goniometer; and (**c**,**d**) the IMU_flex_ and goniometer. Squared correlation coefficients (r^2^) and limits of agreements are presented.

**Figure 6 sensors-22-01690-f006:**
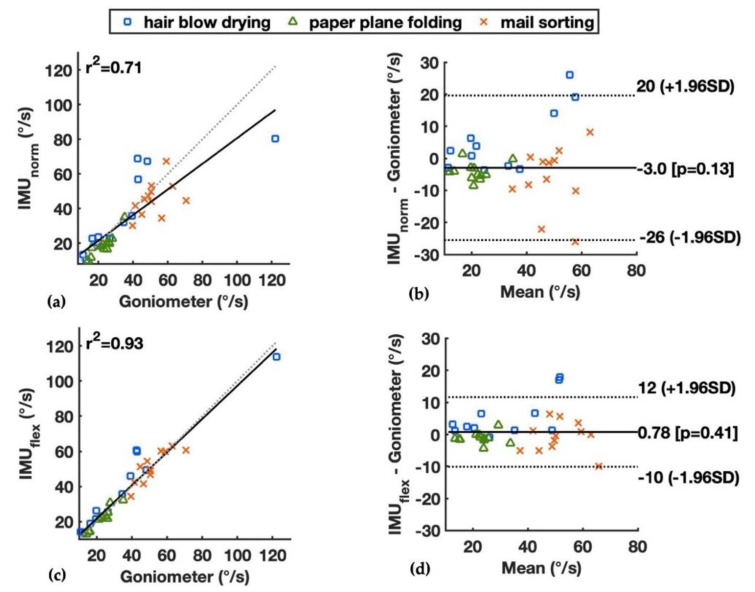
Median wrist flexion velocity for three simulated work tasks measured by the goniometer and the IMU-based methods, showing by linear correlation and Bland–Altman plots: (**a**,**b**) the IMU_norm_ and goniometer; and (**c**,**d**) the IMU_flex_ and goniometer. Squared correlation coefficients (r^2^) and limits of agreements are presented.

**Table 1 sensors-22-01690-t001:** The mean and standard deviation (SD) of wrist flexion velocity from the goniometer, the IMU_norm_ and the IMU_flex_, and the mean absolute errors (MAEs) and SD for the comparisons of methods during the standard wrist movements at paces of 30, 60, and 90 beats per minute (BPM). Data are shown for the 10th, 50th, and 90th percentiles of the wrist velocity with 12 participants included.

°/s	Percentile	Individual Method, Mean (SD)			Comparison of Methods, MAEs (SD)	
		Goniometer	IMU_norm_	IMU_flex_	IMU_norm_−Goniometer	IMU_flex_−Goniometer
Flexion/Extension						
30 BPM	10th	4.2 (3.2)	4.8 (2.2)	4.2 (3.1)	1.4 (0.9)	0.3 (0.2)
	50th	40.7 (17.5)	36.9 (15.5)	39.4 (16.9)	4.1 (2.6)	1.4 (1.2)
	90th	125.7 (52.0)	116.8 (48.5)	118.6 (47.2)	9.0 (4.2)	7.5 (6.4)
60 BPM	10th	4.0 (4.7)	4.3 (3.3)	4.0 (5.1)	1.3 (0.8)	0.4 (0.5)
	50th	72.1 (40.2)	64.8 (37.0)	69.4 (39.7)	7.8 (5.5)	3.1 (2.8)
	90th	233.1 (102.6)	217.9 (98.0)	219.2 (92.3)	15.2 (7.5)	14.3 (12.1)
90 BPM	10th	6.6 (9.2)	5.4 (6.5)	6.8 (10.5)	2.0 (3.0)	1.0 (1.4)
	50th	90.6 (63.7)	81.4 (60.1)	87.8 (63.3)	10.1 (7.7)	4.1 (3.2)
	90th	322.0 (145.7)	300.8 (143.3)	301.8 (136.7)	21.4 (12.1)	20.2 (14.1)
Radial/Ulnar Deviation						
30 BPM	10th	0.5 (0.2)	1.4 (0.6)	0.5 (0.2)	0.9 (0.5)	0.1 (0.1)
	50th	4.2 (1.6)	12.0 (6.1)	3.7 (1.5)	7.9 (5.2)	0.8 (0.7)
	90th	16.9 (7.1)	51.2 (19.2)	16.1 (7.4)	34.3 (16.1)	2.9 (2.9)
60 BPM	10th	0.9 (0.5)	1.7 (0.8)	0.8 (0.3)	0.9 (0.7)	0.2 (0.2)
	50th	7.9 (4.3)	22.3 (13.2)	7.2 (3.4)	14.4 (11.9)	1.4 (1.9)
	90th	37.0 (35.9)	101.2 (35.0)	39.1 (35.7)	65.1 (32.5)	6.8 (7.6)
90 BPM	10th	0.9 (0.7)	2.0 (1.2)	0.9 (0.7)	1.1 (0.7)	0.2 (0.1)
	50th	7.9 (3.7)	23.4 (18.0)	8.1 (4.1)	15.7 (14.9)	1.3 (1.2)
	90th	33.5 (16.0)	130.0 (55.5)	38.5 (22.3)	98.5 (47.9)	9.0 (12.4)
Pronation/Supination						
30 BPM	10th	1.1 (0.5)	1.3 (0.8)	0.7 (0.2)	0.3 (0.3)	0.5 (0.3)
	50th	10.2 (5.0)	26.2 (10.1)	5.0 (1.3)	16.1 (8.1)	5.2 (4.5)
	90th	39.5 (15.4)	77.2 (29.2)	18.9 (6.6)	37.7 (23.8)	20.6 (12.8)
60 BPM	10th	1.6 (1.0)	1.9 (1.9)	1.2 (0.7)	0.7 (1.0)	0.5 (0.4)
	50th	18.5 (10.0)	46.8 (23.1)	10.1 (7.4)	29.1 (19.0)	8.6 (8.3)
	90th	103.6 (100.8)	170.1 (89.6)	61.3 (105.3)	69.7 (49.3)	42.3 (26.7)
90 BPM	10th	1.6 (1.5)	2.0 (2.0)	1.0 (0.7)	0.6 (0.9)	0.7 (0.9)
	50th	19.1 (10.5)	51.4 (32.7)	9.6 (5.0)	32.3 (24.7)	9.5 (6.4)
	90th	101.4 (54.4)	197.3 (83.6)	41.3 (17.2)	96.6 (50.8)	60.1 (45.7)

**Table 2 sensors-22-01690-t002:** The mean and standard deviation (SD) of wrist flexion velocity from the goniometer, the IMU_norm_ and the IMU_flex_, and the mean absolute errors (MAEs) and SD for the comparisons of methods during the simulated work tasks. Data are shown for the 10th, 50th, and 90th percentiles of the wrist velocity during each task with 12 participants included.

°/s	Percentile	Individual Method, Mean (SD)			Comparison of Methods, MAEs (SD)	
		Goniometer	IMU_norm_	IMU_flex_	IMU_norm_−Goniometer	IMU_flex_−Goniometer
Blow-drying hair	10th	5.2 (4.0)	5.8 (4.2)	6.6 (4.5)	1.4 (1.7)	1.4 (1.6)
	50th	36.3 (29.9)	37.8 (24.0)	40.5 (28.4)	10.6 (12.7)	5.8 (6.0)
	90th	111.7 (58.8)	105.9 (44.3)	115.6 (54.9)	24.6 (26.0)	10.8 (10.3)
Folding paper planes	10th	3.1 (1.0)	2.6 (1.0)	2.9 (1.1)	0.6 (0.5)	0.3 (0.2)
	50th	23.1 (5.9)	18.9 (6.2)	22.0 (6.0)	4.4 (2.3)	1.6 (1.2)
	90th	93.0 (18.7)	74.9 (21.9)	92.0 (22.7)	19.1 (9.5)	7.9 (5.1)
Sorting mail	10th	8.9 (1.5)	7.4 (2.4)	8.5 (1.7)	2.2 (1.6)	0.8 (0.4)
	50th	51.8 (9.1)	45.6 (9.8)	51.1 (8.9)	8.1 (8.4)	3.6 (2.9)
	90th	145.4 (19.9)	122.4 (21.8)	140.5 (23.0)	25.8 (13.5)	7.9 (6.5)

## Data Availability

Data presented in the paper are available on request from the corresponding author L.Y.

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
