# Peer review of "Evaluation of a New Simplified Inertial Sensor Method against Electrogoniometer for Measuring Wrist Motion in Occupational Studies"

_sensors, 2022, doi:10.3390/s22041690_

Round 1

Reviewer 1 Report

This study compared a simple IMU-based method with the electrogoniometer for measuring wrist flexion velocity during standard wrist movements and simulated work tasks. As expected, the IMU method had better performance, lower bias in the measurement than the goniometer. There are some comments about this study.

1) The IMU method in this study is not a new method. As the IMUs have been widely used in motion detection, numerous related works can be referenced and discussed. 

2) The signal processing and statistical analysis introduced in Section 2 are relatively simple and trivial. Actually, the gyro has the drift problem, but this study did not address this sort of problems which could affect the accuracy of the results.

3) The results came from only 12 participants which are considered not enough for statistical significance analysis.

4) Over half of the listed references are considered old (early than 2010). 

5) The calibration method should be addressed at this stage, not in the future work. 

Author Response

Thank you for reviewing the manuscript and for providing suggestions on how the manuscript can be further improved. We have answered your comments point-to-point in the following text:

This study compared a simple IMU-based method with the electrogoniometer for measuring wrist flexion velocity during standard wrist movements and simulated work tasks. As expected, the IMU method had better performance, lower bias in the measurement than the goniometer. There are some comments about this study.

1) The IMU method in this study is not a new method. As the IMUs have been widely used in motion detection, numerous related works can be referenced and discussed. 

Thanks for the comment! We have updated the text to include more work on the IMU in motion detection in the introduction. Still, a lot of the IMU systems are too complex to use in the field on a large scale by practitioners, and they mostly include assessment of the upper arm and trunk motion and not many focused on wrist velocity. Therefore, a new tool with better convenience and lower cost is needed, which is the drive of our study.

2) The signal processing and statistical analysis introduced in Section 2 are relatively simple and trivial. Actually, the gyro has the drift problem, but this study did not address this sort of problems which could affect the accuracy of the results.

Thanks for the comment! The strength of our method is actually that the gyro drift does not impact our results, since we are not using the gyro to compensate for accelerometer signals and the noise of gyro won’t be accumulated, which are commonly done in other systems. We have added text in the intro to explain it.

3) The results came from only 12 participants which are considered not enough for statistical significance analysis. 

Thanks for the comment! The choice of number of participants was based on previous studies where wrist motion were measured with technical measurement systems (reference 36, 40), and other validation studies of technical measurement systems of other body parts (references 21, 31). In our results we could also show distinctive differences between the two algorithms for IMU-based method against the goniometer method. We therefore consider 12 participants to be a reasonable sample size for our study.

4) Over half of the listed references are considered old (early than 2010). 

Thanks for the comment. We have included a few more newer references regarding IMU systems in the manuscript. It is also worth mentioning that there are pretty few publications focusing on wrist/hand risk assessment using wearable sensors in the recent decade.

5) The calibration method should be addressed at this stage, not in the future work.

Thanks for the suggestion. We have added explanation in the discussion, as the calibration method would be needed if the IMUs are not mounted aligned meticulously in field studies or by practitioners as their work routine of risk assessment. At the current stage in the laboratory environment, our focus was to test and validate the system in a controlled condition and the two IMUs were placed in alignment with caution. Therefore, we think the calibration method can be explored for field studies. We also didn’t test different calibration procedures in our lab experiment as this was not the focus of this study.

Reviewer 2 Report

Thank you for the opportunity to review this manuscript.
The aim of this study was to evaluate a new simplified IMU-based method, with two computational algorithms, i.e. IMUnorm and IMUflex, against an electrogoniometer for measuring wrist flexion velocity during standard wrist movements and simulated work tasks. 
In situations where practical ergonomic problems have to be solved, simplicity, utility, and face validity of the method are more important than expressing results in exact numeric figures. Validity and repeatability of data are  particularly important in research and when comparing exposures to safety limits. Thus the selection of the most appropriate tool must involve consideration of the analysis’ objectives and how the results will be used – a tool to help identify improvement opportunities may not have the same precision requirements. 
It should be mentioned that IMUs have quite a few limitations that are mentioned by many authors, please mention them in this manuscript
Were appropriate criteria for the inclusion and exclusion of participants applied?
I am sorry but I did not understand the measurement method, why are the IMU sensors attached to the goniomet and not placed directly on the body of the participant?
The participant in photo 1 has additionally reflective markers attached, was the measurement performed with the optometric system at the same time?
I understand that one measurement point was on the head of the third metacarpal bone and the other on the forearm, please provide the exact anatomical location of this point
What were the authors of the selection of activities guided by?
How was the repeatability of performed tasks ensured?

Author Response

Thank you for reviewing the manuscript and for providing suggestions on how the manuscript can be further improved! We have answered your comments point-by-point in the following text:

Reviewer 2.

Thank you for the opportunity to review this manuscript.
The aim of this study was to evaluate a new simplified IMU-based method, with two computational algorithms, i.e. IMUnorm and IMUflex, against an electrogoniometer for measuring wrist flexion velocity during standard wrist movements and simulated work tasks. 
In situations where practical ergonomic problems have to be solved, simplicity, utility, and face validity of the method are more important than expressing results in exact numeric figures. Validity and repeatability of data are  particularly important in research and when comparing exposures to safety limits. Thus the selection of the most appropriate tool must involve consideration of the analysis’ objectives and how the results will be used – a tool to help identify improvement opportunities may not have the same precision requirements. 

It should be mentioned that IMUs have quite a few limitations that are mentioned by many authors, please mention them in this manuscript

Thanks for the comment! The strength of our method is that the gyro drift does not impact our results, since we are not using the gyro to compensate for accelerometer signals and the noise of gyro won’t be accumulated, which are commonly done in other IMU-based systems. We have added text in the intro to explain it.

Were appropriate criteria for the inclusion and exclusion of participants applied?

Thanks for the question. Participants included were above the age of 18, healthy and free from pain in their dominant hand. Participants were questioned for symptoms of COVID and infections ahead of the study. Additional information is added to the text.

I am sorry but I did not understand the measurement method, why are the IMU sensors attached to the goniometer and not placed directly on the body of the participant?  

Thanks for the question. In order to equip both systems at the same time for comparison, there was not enough space to put IMU sensors directly on the skin, which would be the idea situation when measuring with IMU sensors alone. Therefore, the IMUs were mounted on the goniometer endblock. We have included the reasoning to our subheading 4.1, paragraph two of our discussion.

The participant in photo 1 has additionally reflective markers attached, was the measurement performed with the optometric system at the same time?

In the experiments, we have used the IMU, goniometer and the optical tracking system. The optical system was traditionally considered as the golden standard for motion tracking in a lab environment. However, we also encountered significant estimation errors with the built-in model of the optical system when it comes to hand/wrist motion, especially during supination. Therefore, we’ve decided to first focus on the comparison of the IMUs and the goniometer system, which is considered as the standard measure for occupational field studies. In a following manuscript, we plan to explore and improve the model of the optical system, and the performance of both the goniometer and the IMUs will be evaluated against the optical system.

I understand that one measurement point was on the head of the third metacarpal bone and the other on the forearm, please provide the exact anatomical location of this point.

Thank you of the suggestion! we have added explanation of the location in the text 2.2. The location was individually adjusted after first placing the hand endblock, so that “the cables between the two endblocks were just kept straight and not squeezed when the wrist was fully extended”

What were the authors of the selection of activities guided by?

The choice of the tasks was intended to represent hand/wrist motion at different intensities, with the possibility to replicate easily in a laboratory environment. The task of folding paper plane was chosen as it had lower intensity in hand motion, and it could be similar to some assembly tasks that requires precision and slow movement.

The tasks of mail sorting and blow drying were chosen as they had higher intensity and also they had been used in previous study for validation sensor-based motion measuring system, e.g. as in L. Yang, W. J. A. Grooten, and M. Forsman, “An iPhone application for upper arm posture and movement measurements,” Appl. Ergon., vol. 65, pp. 492–500, Nov. 2017.

How was the repeatability of performed tasks ensured?

The standard wrist movements were guided by metronome at three paces, which were constrained movements and had a high repeatability. For the simulated work tasks, participants were informed to perform at their own pace and the repeatability of tasks was not aimed for. Since workstyle can be diverse between people in real workplaces, we consider that it is good to have this variation of how participants carry out the work tasks in this study.

Round 2

Reviewer 1 Report

Thanks for considering the previous reviewing comments. However, only a few descriptions and related works have been added in the introduction; neither the designed method nor the sample size of the test have been enhanced in this revised version. 

Author Response

Thank you for providing your review! We have responded and made changes based on the previous review. Six new references 17-22 (one of them is a literature overview on the new wearable systems for occupational ergonomics assessment) have been added to the introduction regarding the recent development of IMU systems. In addition, we have added reference 33-34 to emphasize the integration of gyroscope in many IMU systems and the drift problem it can bring, which our system doesn’t suffer from due to that our algorithms does not integrate gyroscope signals.

Regarding the sample size, we have added references (21, 31, 36, 40) of previous studies showing that the sample size is reasonable for this type of research. We cannot reperform the experiment with more participants unfortunately.

When it comes to the method, can the reviewer clarify what type of new method can be used in this study, based on our current results to improve the quality? If it is the calibration method the reviewer is referring to, we have added explanation in the discussion, as the calibration method would be needed if the IMUs are not mounted aligned meticulously in field studies or by practitioners as their work routine of risk assessment. We aim to explore it in the future study when we perform field experiment. 

Reviewer 2 Report

Thank you for the authors' comments, I am satisfied.

Author Response

Thank you for providing your review and suggestions!